# Self-Reported Periodontal Disease and Its Association with SARS-CoV-2 Infection

**DOI:** 10.3390/ijerph191610306

**Published:** 2022-08-18

**Authors:** Israel Guardado-Luevanos, Ronell Bologna-Molina, José Sergio Zepeda-Nuño, Mario Isiordia-Espinoza, Nelly Molina-Frechero, Rogelio González-González, Mauricio Pérez-Pérez, Sandra López-Verdín

**Affiliations:** 1Postgraduate in Periodontology and Implant Dentistry, Faculty of Dentistry, Autonomous University of Guadalajara, Guadalajara 44100, Mexico; 2Molecular Pathology Area, Faculty of Dentistry, Montevideo Republic University, Montevideo 11200, Uruguay; 3Research Department, School of Dentistry, Juarez University of the State of Durango, Durango 34100, Mexico; 4Microbiology and Pathology Department, Pathology Laboratory, University Center of Health Sciences, University of Guadalajara, Guadalajara 44100, Mexico; 5Clinical Department, Biomedical Science Division, Los Altos University Center, University of Guadalajara, Guadalajara 44100, Mexico; 6Health Care Department, Autonomous Metropolitan University, Mexico City 14387, Mexico; 7Periodontology Postgraduate, Comprehensive Dental Clinics Department, University Center of Health Sciences, University of Guadalajara, Guadalajara 44100, Mexico; 8Research Institute of Dentistry, Comprehensive Dental Clinics Department, University Center of Health Sciences, University of Guadalajara, Guadalajara 44100, Mexico

**Keywords:** SARS-CoV-2, self-report, periodontal disease, case-control studies, COVID-19

## Abstract

**Introduction**: Knowledge of the oral manifestations associated with SARS-CoV-2 infection, the new coronavirus causing the COVID-19 pandemic, was hindered due to the restrictions issued to avoid proximity between people and to stop the rapid spread of the disease, which ultimately results in a hyperinflammatory cytokine storm that can cause death. Because periodontal disease is one of the most frequent inflammatory diseases of the oral cavity, various theories have emerged postulating periodontal disease as a risk factor for developing severe complications associated with COVID-19. This motivated various studies to integrate questions related to periodontal status. For the present work, we used a previously validated self-report, which is a useful tool for facilitating epidemiological studies of periodontal disease on a large scale. **Methodology**: A blinded case-control study with participants matched 1:1 by mean age (37.7 years), sex, tobacco habits and diseases was conducted. After the diagnostic samples for SARS-CoV-2 detection were taken in an ad hoc location at Guadalajara University, the subjects were interviewed using structured questionnaires to gather demographic, epidemiological and COVID-19 symptom information. The self-reported periodontal disease (Self-RPD) questionnaire included six questions, and subjects who met the criteria with a score ≥ 2 were considered to have periodontal disease. **Results**: In total, 369 participants were recruited, with 117 participants included in each group. After indicating the subjects who had self-reported periodontal disease, a statistically significant difference (*p* value ≤ 0.001) was observed, showing that self-reported periodontal disease (n = 95, 85.1%) was higher in SARS-CoV-2-positive individuals than in controls (n = 66, 56.4%), with an OR of 3.3 (1.8–6.0) for SARS-CoV-2 infection in people with self-reported periodontal disease. Cases reported a statistically higher median of symptoms (median = 7.0, Q_1_= 5.5, Q_3_ = 10.0) than controls (*p* value ≤ 0.01), and cases with positive self-RPD had a significantly (*p* value ≤ 0.05) higher number of symptoms (median = 8.0, Q_1_ = 6.0, Q_3_ = 10.0) in comparison with those who did negative self-RPD (median = 6.0, Q_1_ = 5.0, Q_3_ = 8.0). **Conclusions**: According to this study, self-reported periodontal disease could be considered a risk factor for SARS-CoV-2 infection, and these individuals present more symptoms.

## 1. Introduction

At the end of December 2019, a novel coronavirus that was tentatively named severe acute respiratory syndrome (SARS-CoV-2) in Wuhan, a central city in China, was announced by the World Health Organization [1]. SARS-CoV-2 is an RNA virus that has become a major public health concern [2] because it can cause the disease known as coronavirus disease 19 (COVID-19). Individuals with this disease can go through various stages [3], ranging from asymptomatic to severe illness.

The primary entry of the virus occurs through projected droplets, leading to the first contact and colonization of cells in the oral cavity, nose or eyes [4]. The immune system represents an effective barrier against multiple antigens, including viruses; nevertheless, some diseases and conditions can cause immune dysfunction or immunosuppression. In this sense, SARS-CoV-2 may be able to more easily infect tissues altered by inflammatory processes that promote its virulence.

Periodontal disease is a noncontagious chronic disease that is more prevalent in the oral cavity and has been associated with several systemic conditions, including adverse pregnancy outcomes, cardiovascular diseases, type 2 diabetes mellitus, respiratory disorders, fatal pneumonia, chronic renal disease and metabolic syndrome [5].

Thus, periodontal tissues are the best candidates for SARS-CoV-2 infection in the oral cavity. First, the virus can enter because angiotensin-converting enzyme 2 (ACE2) receptor expression has been previously reported in human gingival fibroblast homogenates [6], and Huang et al. recently demonstrated relatively higher expression of ACE2 and TMPRSS2 in the gingival epithelium. These cell populations were included among oral barrier tissues to provide multiple available targets for viral infections [7]. Gingival inflammation can lead to chronic periodontitis, and this condition is promoted by plaque biofilms that contain complex and evolving microbiota nesting. Lamont et al. proposed the “Polymicrobial Synergy and Dysbiosis Model”, wherein inflammation is stimulated in the periodontal pocket, with the subsequent downregulation of host immunity via the subversion of complement, manipulation of neutrophils and the inhibition of macrophage responses [8], which are also required for SARS-CoV-2 virulence [9]. Finally, the presence of different viruses (e.g., HSV, EBV, HCMV) in periodontal pockets has been described [10].

According to the information described above, periodontitis may be a risk factor for COVID-19. However, dental service was forced to close during the acute period of the COVID-19 pandemic due to various reasons, one of which was a lack of personal protective equipment. Additionally, the key factors influencing the possibility of SARS-CoV-2 infection during a dental visit were aerosol-generating procedures, including ultrasonic scaling, which has an impact on decreasing periodontal procedures. This way, reopening of dental services during the pandemic was only to attend to urgent procedures, postponed initial or periodic oral examination and routine dental cleaning, making clinical periodontal evaluation impossible [11,12,13].

Therefore, this study aimed to measure periodontal status through a previously validated test in individuals who were tested for SARS-CoV-2 infection.

## 2. Materials and Methods

A case-control study was conducted between December 2020 and January 2021 in individuals aged 20 to 75 years who met the inclusion criteria for a COVID-19 diagnosis.

Appropriate ethical and biosecurity conduct was ensured by the Declaration of Helsinki guidelines. Study group recruitment was conducted under consent and authorization from the Ethical Committee of the “Center for Multidisciplinary Research in Health” at the University of Guadalajara (CEI-2020-10).

### 2.1. Subjects and Self-Report Questionnaire

In the “COVID-19 diagnostic center” at the University Center-Tonalá, Guadalajara University, the subjects were interviewed using structured questionnaires to gather demographic and epidemiological data, including the following: residence, age and sex of the patient; travel inside or outside of the country in the past 15 days; and the presence of diabetes or hypertension. To be considered for the RT–qPCR for SARS-CoV-2 and be included in the study, presentations of any symptoms related to mild illness, such as headache, vomiting, diarrhea, fever, sore throat, runny nose or other symptoms, in the past 15 days were necessary. Additionally, participants had to know the type of medications they were taking. Exclusion criteria were pregnancy, anticoagulant medication, blood diseases, such as hemophilia, or autoimmune disease, such as lupus.

The diagnostic sample was taken with two swabs, a nasopharyngeal swab and an oropharyngeal swab, by trained medical personnel. RT–qPCR SARS-CoV-2 detection was performed using “DeCoV19Kit Triplex” (GENES2LIFE) reagents, which are designed to detect three regions of the virus nucleocapsid gene (N1, N2 and N3).

No prior knowledge of infection status (one blind) or the self-report periodontal disease (Self-RPD) questionnaire was applied. This test was previously validated by Khader et al. and includes six questions, and subjects who met the criteria with a score ≥ 2 were considered to have positive (+) self-reported periodontal disease [14].

### 2.2. Case-Control Design

To obtain the RT–qPCR SARS-CoV-2 detection results, the participants were included in one of two study groups:(1)The control group included subjects who tested negative for the three genes evaluated for the detection of SARS-CoV-2.(2)The COVID-19 group included subjects who tested positive for two of the three genes evaluated for the detection of SARS-CoV-2.

The study subjects were then matched 1:1 and according to the previously determined associated factors of mean age, sex, smoking habits and systemic diseases. The descriptive data for these variables can be seen in Table 1.

### 2.3. Statistical Analysis

Data were analyzed with the SPSS version 20 (IBM Corp., Armonk, NY, USA) statistical package database. We compared categorical variables using the χ^2^ test or Fisher’s exact test, and unadjusted odds ratios (ORs) with 95% confidence intervals (CIs) were calculated. The Shapiro–Wilk test was used to determine the normality distribution of the numerical variables. The analysis was mainly descriptive. We considered a *p* value lower than 0.05 to be statistically significant.

## 3. Results

A total of 229 participants who tested positive for SARS-CoV-2 and 140 participants who tested negative for SARS-CoV-2 were recruited. For matching purposes, 117 assistants were included in each group. At the time of matching, the mean age was 37.7 years; according to sex, 69 (59.0%) females and 48 (41.0%) males were selected for each group; 30 (25.6) people who smoked and 87 (74.4) who did not smoke were included; and the last match took the presence of diseases, such as diabetes and hypertension (n = 22, 18.8%), or the absence of any diagnosis of disease (n = 95, 81.2%) into account.

The distribution of unmatched characteristics can be seen in Table 1. Drug intake was higher in the case group (n = 61, 52.1%) than in the control group (n = 39, 33.3%). Of the 22 symptoms registered, 5 showed statistically significant differences between the case and control groups, where headache (*p* value < 0.05), flu (*p* value < 0.05), conjunctivitis (*p* value < 0.05), olfactory disturbance (*p* values < 0.001) and gustatory disturbance (*p* values < 0.05) were more frequent in the case group (Table 1).

### 3.1. Self-Reported Periodontal Disease Differences between the Case and Control Groups

Six questions were created based on a previously validated test. When we analyzed the differences between the answers to each question by group, even those that did not show significant differences, we observed a tendency for one particular question, “Do you have impaction between your teeth?”, with the answers of “yes” being more frequent in COVID-19-positive individuals (n = 64, 54.7%) (*p* value > 0.05) (Table 2). After indicating the subjects who had self-reported periodontal disease and analyzing by the chi square test, a statistically significant difference (*p* value < 0.001) was observed, showing that positive self-RPD (n = 95, 85.1%) was often higher in SARS-CoV-2-positive individuals than in the controls (n = 66, 56.4%), with an OR of 3.3 (1.8–6.0) for SARS-CoV-2 infection in people with self-reported periodontal disease (Figure 1).

### 3.2. Self-Reported Periodontal Disease in SARS-CoV-2-Infected Patients and Their Symptoms

We hypothesized that COVID-19-positive patients with self-reported periodontal disease were predisposed to have more symptoms than those who did not. We used a nonparametric test to analyze the number of symptoms, and the normality distribution statistical test was significant (*p* = 0.012). First, we contrasted the differences between the case and control groups and found that cases presented a significantly higher median of symptoms (median = 7.0, Q_1_= 5.5, Q_3_ = 10.0) than the controls (median = 6.0, Q_1_ = 4.0, Q_3_ = 8.5) (*p* value < 0.01). Second, independently in the case and control groups, we observed that cases with self-reported periodontal disease had a significantly (*p *value < 0.05) higher number of symptoms (median = 8.0, Q_1_ = 6.0, Q_3_ = 10) than those who did not (median = 6.0, Q_1_ = 5.0, Q_3_ = 8.0). Even in cases with positive self-RPD, individuals had fewer than two symptoms as a minimum. However, these differences were not present in the control group (Figure 2).

## 4. Discussion

In this pandemic, studies have focused on understanding the behavior of the SARS-CoV-2 virus and COVID-19. COVID-19 affects people in different ways, with patients exhibiting a range of symptoms and severities. While risk factors, such as age, sex and comorbidities, have been highlighted as increasing the risk of complications and mortality, there is still a high proportion of patients with no identified risk factors who suffer from severe side effects and complications. As many as 10–15% of people under 60 years old with no risk factors exhibit moderate to severe reactions to COVID-19 [1].

Periodontal disease is one of the most prevalent oral diseases, and an inflammatory nature has been implicated in multiple conditions. Since the beginning of the pandemic, Pitones-Rubio et al. have summarized enough evidence to propose that periodontal disease acts as a risk factor for COVID-19 due to the previous implications with other comorbidities, such as diabetes, as well as the promotion of oral dysbiosis [15].

Several mechanisms have been proposed to explain the involvement of periodontal disease and its ability to potentiate the severity of COVID-19 [16], including the aspiration of periodontal pathogenic microbiota, which represents a powerful proinflammatory stimulus or by inducing ACE2 overexpression, with a subsequent increase in IL-6 and IL-8 cytokines by bronchial cells and alveolar epithelial cells [17]. Th17 cells play a central role during the establishment of periodontal disease, and these cells regulate the expression of proinflammatory mediators, such as IL6, IL-17 and IL-23 [18]. In this sense, overexpression of the Th17 pathway has been observed in patients with SARS-CoV-2, and this Th17 inflammatory profile has been implicated in the development of the cytokine storm, as well as lung disease, edema and tissue damage in lung infections, including those caused by SARS-CoV-2 [19]. The existence of these inflammatory pathways in common could represent an association between periodontal disease as a predisposing factor for the development of adverse effects related to COVID-19.

Until now, the pandemic has made it impossible to evaluate the clinical indicator of periodontal disease and create a reliable self-reported test to explore its relationship with COVID-19. According to Basso et al., following a review of seven original papers, only one work evaluated the association between periodontitis and COVID-19 [20].

For example, Larvin et al. found that self-reported painful gums, bleeding gums and loose teeth in COVID-19-positive participants resulted in significantly higher mortality for participants with periodontal disease [21]. However, contrary to our findings, the risk of COVID-19 infection in participants with self-reported painful/bleeding or loose teeth was not increased compared to the controls. This could have been due to the difference in mean age between the studies: subjects in the study by Larvin et al. had a mean age of 67 years, and they probably included subjects with more loose teeth, which acted as a confounding variable. Another reason could be that the indicators were analyzed independently. An advantage of the present study is that it considered self-reported periodontal disease based on a score, in which more than one affirmative answer was necessary.

Marouf et al., who assessed periodontal conditions using dental radiographs from the Qatar state database, which includes 14 hospitals, found that periodontitis was associated with COVID-19 complications (intensive care unit admission and the need for assisted ventilation), including death [22].

This is the first study to evaluate a cohort that was exposed to SARS-CoV-2, completely blinded to the knowledge of the diagnosis of SARS-CoV-2 infection at the time of the intervention, and we observed that people with self-reported periodontal disease were at high risk of developing SARS-CoV-2 infection and had a greater number of symptoms at the onset of illness.

Simultaneously, during the pandemic period, various studies used a self-report survey with the aim of describing the oral manifestations associated with COVID-19 [23,24,25,26,27]. Xerostomia, altered taste and burning mouth were considered the most frequent symptoms. Gum bleeding was reported in two studies at a frequency range of 7–17.5% [28]. Although Biadsee et al. mentioned that oral hygiene did not contribute to the presence of these three symptoms [24], we cannot ensure that these symptoms do not exacerbate periodontal symptoms in the population included in the present study.

More symptomatology in SARS-CoV-2-positive patients with self-reported periodontal disease in the present study could be in concordance with Abubakr et al., who found that patients with poor oral hygiene had oral manifestations, such as ulcerations and oral/dental pain, more often than those with good oral hygiene [23]. These similarities could show the impact of periodontal disease on the immune system by contributing to bacterial superinfection, a common event in severe cases of COVID-19, which exhausts cells, reducing lymphocyte levels. This must also be validated with more precise clinical and molecular parameters.

The presence of a periodontal pocket should be detected in patients infected with SARS-CoV-2 because SARS-CoV-2 RNA has been detected in periodontal tissues [29], making it possible that people with periodontal disease are potential vectors of transmission. There may therefore be an increased risk of suffering more complications from COVID-19 disease, simultaneously increasing the risk of death. Finally, there may be an association of osteonecrosis in people who suffered from COVID-19, an oral pathology strongly conditioned by the periodontal status [30].

Applying a self-reported periodontal disease in communities is easier than applying periodontal clinical evaluations as it is less expensive and can be performed by any health personnel; thus, it could be used as an adjuvant tool in the anamnesis of patients with suspected SARS-CoV-2 infection.

According to this work, people should be informed about oral hygiene using this self-report questionnaire. However, due to the subjectivity involved in this type of questionnaire, they must be attended by a professional to determine the degree of periodontal disease.

The main limitation of this study was the translation of the questions from English to Spanish, as there could have been interpretation errors. Additionally, we did not know the level of schooling from the respondents, which could have compromised the understanding of the question.

## 5. Conclusions

In conclusion, self-reported periodontal disease can be an adjuvant marker to assume the risk of SARS-CoV-2 infection, and these individuals present more symptoms at the onset of the disease.

## Figures and Tables

**Figure 1 ijerph-19-10306-f001:**
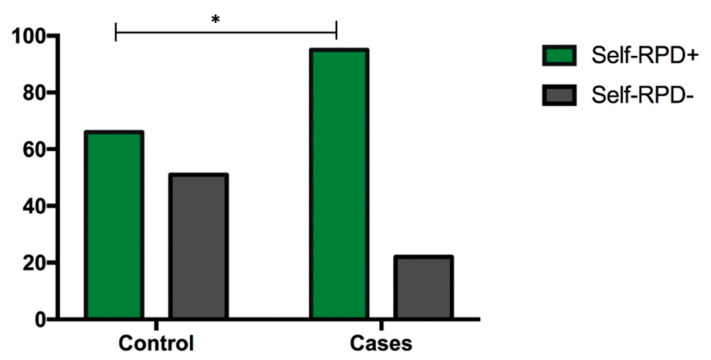
Self-RPD between the case and control groups. Control self-RPD positive (+): n = 66, 54.4%; Control self-RPD negative (−): n = 51,43.6%, Cases self-RPD positive (+): n = 95, 81.2%; Cases self-RPD negative (−).: n = 22, 18.8%. Statistics: χ^2^ test. Abbreviations and symbols: Self-RPD: Self-reported periodontitis disease, * *p* value < 0.001.

**Figure 2 ijerph-19-10306-f002:**
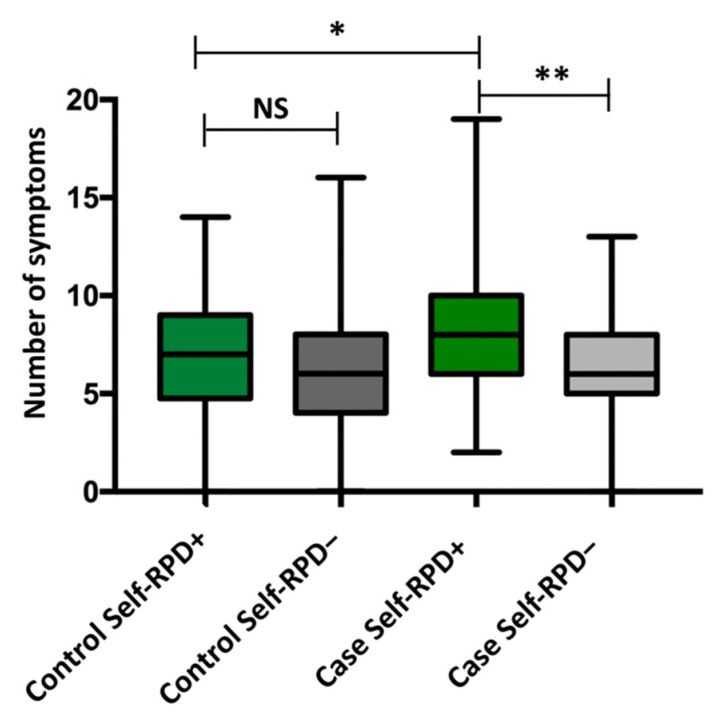
Self-RPD and the number of symptoms between groups. Control self-RPD positive (+): median = 7.0, Q_1_ = 4.7, Q_3_ = 9.0, minimum = 0.0, maximum = 14.0; Control self-RPD negative (−): median = 6.0, Q_1_ = 4.0, Q_3_ = 8.0, minimum = 0.0, maximum = 16.0; Case self-RPD positive (+): median = 8.0, Q_1_ = 6.0, Q_3_ = 10.0, minimum = 2.0, maximum = 19.0; Case self-RPD negative (−): median= 6.0, Q_1_ = 5.0, Q_3_ = 7.0, minimum = 0.0, maximum = 13.0. Statistics: Mann–Whitney U. Abbreviations and symbols: Self-RPD: Self-reported periodontitis disease, NS: Not significant, * *p* value < 0.01, ** *p* value < 0.05.

**Table 1 ijerph-19-10306-t001:** Case-control descriptive data and *p* values.

	Controls (n = 117)	Cases (n = 117)	*p*-Value
*MATCHED*			
**Age, mean SD, min, max**	37.7 ± 13.2, 17–67 years.	37.7 ± 12.5, 17–66 years.	1.00
**Sex**			
Female	69 (59)	69 (59)	1.00
Male	48 (41)	48 (41)	
**Smoking**			
Yes	30 (25.6)	30 (25.6)	1.00
No	87 (74.4)	87(74.4)	
**Diseases**			
Yes	22 (18.8)	22 (18.8)	1.00
No	95 (81.2)	95 (81.2)	
*UNMATCHED*			
**Drugs**			
Yes	39 (33.3)	61 (52.1)	0.00 *
No	78 (66.7)	56 (47.9)	
**Symptoms**			
Fever	58 (51.3)	59 (51.3)	1.00
Dry cough	71 (67)	76 (68.5)	0.88
Nasal congestion	19 (19.4)	24 (24.2)	0.01
Tiredness	57 (54.3)	73 (64)	0.16
Coughing up phlegm	14 (14.4)	15 (15.2)	1.00
Shortness of breath	17 (17.5)	15 (15)	0.70
Cut Body	43 (42.6)	35 (33.7)	0.19
Headache	78 (70.9)	98 (82.5)	0.01*
Chills	34 (34.0)	35 (33.3)	1.00
Muscle pain	44 (43.1)	51 (46.8)	0.67
Joint pain	29 (29.6)	40 (37.4)	0.30
Runny nose	34 (33.3)	35 (33.3)	1.00
Burning throat	53 (50.5)	57 (53.3)	0.39
Flu	30 (30.3)	48 (46.2)	0.02 *
Conjunctivitis	17 (17.5)	33 (32.0)	0.02 *
Diarrhea	22 (22.4)	27 (26.0)	0.62
Vomit	13 (13.5)	7 (7.1)	0.16
Stomach ache	18 (18.6)	11 (11.1)	0.16
Fast breathing	8 (8.6)	5 (5.2)	0.39
Convulsions	1 (1.1)	1 (1.0)	1.00
Olfactory disturbance	35 (35.0)	59 (57.8)	0.00 *
Gustatory disturbance	32 (32.3)	51 (49.0)	0.01 *

Frequencies were based solely on participants included for match purposes. Cases: Positive for SARS-CoV-2; Controls: Negative for SARS-CoV-2, χ^2^ test, * *p* < 0.05, 95% CI.

**Table 2 ijerph-19-10306-t002:** Questions of self-reported periodontal disease differences between cases and control: descriptive data and *p* values.

	Controls (n = 117)	Cases (n = 117)	*p*-Value
	n	%	n	%	
** *Do you have periodontal disease or gum disease?* **	
Yes	42	35.9	56	47.9	0.08
No	75	64.1	61	52.1	
** *Have you ever been told by a dentist that you have periodontal/gum disease with bone loss?* **	
Yes	16	13.7	22	18.8	0.37
No	101	86.3	95	81.2	
** *Have you found any area that is redder than it should be?* **	
Yes	31	26.5	32	27.4	1.00
No	86	73.5	85	72.6	
** *Do your teeth move?* **	
Yes	9	7.7	5	4.3	0.40
No	108	92.3	112	95.7	
** *Are your teeth impacted?* **	
Yes	49	41.9	64	54.7	0.06
No	68	58.1	53	45.3	
** *Do you notice that your teeth are getting longer?* **	
Yes	7	6.0	6	5.0	0.50
No	110	94.0	111	95.0

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
