# Peer review of "Self-Reported Periodontal Disease and Its Association with SARS-CoV-2 Infection"

_ijerph, 2022, doi:10.3390/ijerph191610306_

Round 1

Reviewer 1 Report

The Authors have partly referred to the comments of the Reviewers and improved the manuscript. However, further revisions are necessary.

In the Introduction, it is recommended to add a paragraph on restrictions on access to dental care (including periodontal procedures) during the first wave of the pandemic with appropriate references (e. g. doi: 10.3390/ijerph18157963, 10.3390/ijerph18073421, 10.3390/ijerph17093325). Also, it is suggested to add more articles from 2021-2022 to the references in the Discussion, considering oral manifestations during SARS-CoV-2 infection, especially xerostomia (e. g. doi: 10.3390/jcm9103218, 10.3390/jcm11082202, 10.3390/ijerph182312511). Previously, the Authors have chosen only a first suggested reference, but their manuscript should also be complemented by others.

Moreover, the article requires considerable editorial and language editing – e. g.:

In the case of p-values, „<” should be instead of „≤”;

“Figure” should be instead of “Image”.

Please rephrase this sentence: “Because Shapiro-Wilk was significant (p= 0.012)…”.

Unfortunately, the Authors did not use the template to submit the manuscript. The lack of line numbering makes it very difficult to perform a review.

Reviewer 2 Report

Please revise.

1. Abstract- the sentences in the abstract are not correctly constructed. Please revise looking to similar papers of self-reported periodontitis disease in cardio or other diseases

2. Discussion still looks very superficial. The authors can explore common inflammatory mediators, and common genetic pathways and make the discussion more comprehensive

Author Response

This manuscript is a resubmission of an earlier submission. The following is a list of the peer review reports and author responses from that submission.

Round 1

Reviewer 1 Report

There is a major flow.

The authors say the OR is 3.3 (1.849 – 6.022) in description.

However according to table 2 the OR is 1.6 (.9–2.7) and p value is 0.08. So not significant

Figure 1 is not based on data shown in table 1 where they have reported number of gum diseases in covid positive and covid negative cases

The results says-

"A total of 229 participants who tested positive for SARS-CoV-2 and 140 participants 121 who tested negative for SARS-CoV-2 were recruited. To match the purpose, 117 assistants were included in each group"

However table 1 says a total of 234 (117+117) is recruited.

Reviewer 2 Report

The topic of the manuscript is to evaluate if it is possible that periodontal disease could be a risk factor for SARS-CoV-2 infection based on the self-reported questionnaire among patients from the COVID-19 diagnostic centre in CU-Tonala, Guadalajara University.

The title and the abstract of the article are informative. The Introduction presents the issue of periodontal disease, especially its potential connection with SARS-CoV-2 infection. The section "Materials and Methods" briefly explains the chosen study design. The section "Results" should be improved due to improper presentation of the obtained data. The Discussion is interestingly written, however, should be supplemented with more references and the study limitations. The Conclusions seem to be the "take-home" messages but should be extracted from the Discussion.

Some following points must be clarified/corrected for the further processing of this article.

Merits-related comments:

  1. Please complete keywords with the proper MeSH terms necessary for indexing in the databases.
  2. Please change the objective of the study to make it more reader-friendly and self-explanatory (without reference).
  3. Please explain how the selected control group was matched with the study group. It is not entirely clear how both groups were selected from all participants.
  4. It is appropriate to specify the inclusion and exclusion criteria for the study participants.
  5. There is no information on the test used to investigate the normality of the variable distribution. Did the distribution of all presented numerical results correspond to the normal distribution? If not, these results should be presented in the form of medians and quartiles (instead of means and standard deviation). Even then non-parametric tests should be used.
  6. In the Introduction, it is recommended to add a paragraph on restrictions on access to dental care during the first wave of the pandemic with appropriate references (e. g. doi: 10.3390/ijerph18157963, 10.3390/ijerph18073421, 10.3390/ijerph17093325).
  7. Also, it is suggested to add more articles from 2021-2022 to the references in the Discussion, considering oral manifestations during SARS-CoV-2 infection (e. g. doi: 10.3390/jcm9103218, 10.3390/jcm11082202, 10.3390/ijerph182312511).
  8. At the end of the Discussion, in the new paragraph, the potential limitations of the study should be explained.

Technical comments:

  1. The article also requires considerable editorial editing (table, values, references). Please check the main text and all tables with results carefully – e. g. some zeros are omitted before decimal characters. All abbreviations included in the tables should be explained in the legends below.
  2. The section “Methodology” should be “Materials and Methods”. Also, a separate section with Conclusions is proposed.
  3. Phrases "p-value" should be used instead of "p". Also, “p-value <0.001” instead “p=.000”.
  4. The size of the fonts in Figure 1 should be larger. In its current form, the text is unreadable. Diagrams b and c from Figure 1 should be separated as Figure 2.
  5. In the text, reference numbers should be placed in square brackets [ ], and placed before the punctuation; for example [1], [1–3] or [1,3].
  6. All acronyms or abbreviations (e. g. COVID-19, ACE2) should be defined the first time they appear in each of three sections: the abstract; the main text; the first figure or table. When defined for the first time, they should be added in parentheses after the written-out form.
  7. The citation list must be corrected. References should be described as follows:
  8. 1. Author 1, A.B.; Author 2, C.D. Title of the article. Abbreviated Journal Name Year, Volume, page range.
  9. In Author Contributions, the following statements should be used "Conceptualization, X.X. and Y.Y.; Methodology, X.X.; Software, X.X.; Validation, X.X., Y.Y. and Z.Z.; Formal Analysis, X.X.; Investigation, X.X.; Resources, X.X.; Data Curation, X.X.; Writing – Original Draft Preparation, X.X.; Writing – Review & Editing, X.X.; Visualization, X.X.; Supervision, X.X.; Project Administration, X.X.; Funding Acquisition, Y.Y.".
  10. Also, missing obligatory sections such as Funding, Institutional Review Board Statement, Informed Consent Statement, Data Availability Statement, Conflicts of Interest.
  11. The full names of the authors should be included under the title of the manuscript. The affiliations should be translated into English.